

# Thermal Annealing of Implanted [252]Cf Fission-Tracks in Monazite
**Sean Jones, Andrew Gleadow, Barry Kohn**
**School of Earth Sciences, University of Melbourne, Victoria 3010, Australia**
**Correspondence:** Sean Jones (seanj1@student.unimelb.edu.au)
***Abstract***
A series of isochronal heating experiments were performed to constrain monazite fission-
track thermal annealing properties. [252]Cf fission-tracks were implanted into monazite crystals
from the Devonian Harcourt Granodiorite (Victoria, Australia) on polished surfaces oriented
parallel and perpendicular to (100) prismatic faces. Tracks were annealed over 1, 10, 100 and
1000 hour schedules at temperatures between 30°C and 400°C. Track lengths were measured
on captured digital image stacks, and then converted to calculated mean lengths of equivalent
confined fission tracks which progressively decreased with increasing temperature and time.
Annealing is anisotropic, with tracks on surfaces perpendicular to the crystallographic c-axis
consistently annealing faster than those on surfaces parallel to c. To investigate how the mean
track lengths decreased as a function of annealing time and temperature, one parallel and
two fanning models were fitted to the empirical dataset. The temperature limits of the
monazite partial annealing zone (MPAZ) were defined as length reductions to 0.95 (lowest)
and 0.5 (highest) for this study. Extrapolation of the laboratory experiments to geological
timescales indicates that for a heating duration of $10^7$ years, estimated temperature ranges
of the MPAZ are -44 to 101°C for the parallel model and -71 to 143°C (both $\pm$ 6 - 21°C, 2
standard errors) for the best fitting linear fanning model ($T_0 = \infty$). If a monazite fission-track
closure temperature is approximated as the mid-point of the MPAZ, these results, for tracks
with similar mass and energy distributions to those involved in spontaneous fission of [238]U,
are consistent with previously estimated closure temperatures (calculated from substantially
higher energy particles) of <50°C and perhaps not much above ambient surface
temperatures. Based on our findings we estimate that this closure temperature ($T_c$) for fission



tracks in monazite ranges between ~45 and 25°C over geological timescales of $10^6 - 10^7$ years
making this system potentially useful as an ultra-low temperature thermochronometer.

## 1. Introduction

Fission track thermochronology is an analytical technique used to reconstruct the low-
temperature thermal history of rocks over geological time. Fission tracks form from the
spontaneous nuclear fission of $^{238}$U, resulting in the accumulation of narrow damage trails in
uranium-bearing minerals such as apatite and zircon. The time since the fission tracks began
to accumulate may be calculated by measuring the spontaneous track density and uranium
concentration. If the host rock experienced elevated temperatures, the fission tracks that
have formed up to that point will progressively anneal and eventually disappear. Thermal
diffusion drives the annealing process, with the reduction in fission track density and confined
track length being a function of heating time and temperature in the host rock. From the
apparent age and track length distribution a quantitative analysis of the thermal history of
the host rock can be achieved. For fundamentals of the fission track technique, including
methodology and applications see Wagner and Van den Haute (1992) and Malusa and
Fitzgerald (2019).

The occurrence of monazite as an accessory mineral, along with the presence of significant
uranium (U) and thorium (Th) incorporated in its crystal lattice make it a useful mineral for
isotopic and chemical dating (e.g. Badr et al., 2010; Cenki-Tok et al., 2016; Tickyj et al., 2004).
In monazite, studies have mostly focused on the U-Th-Pb and (U-Th)/He systems but only
limited research has been carried out into the potential of the fission track system, mainly
due to technological limitations. Conventional fission track dating relies on thermal neutron
irradiation of samples to obtain an estimate of $^{238}$U content via the formation of $^{235}$U fission
tracks, usually captured in an adjacent external solid-state track detector such as mica. This
approach, however, has hindered the development of monazite fission track dating for a
number of reasons. Monazite is highly unsuitable for irradiation due to massive self-shielding
by thermal neutron capture from gadolinium (Gd), which may reach abundances in excess of
2 wt%. Gd has an extremely high thermal neutron capture cross-section of 48,890 barns,
averaged over its constituent isotopes, compared to 580 barns for $^{235}$U fission (Gleadow et
al., 2004; Weise et al., 2009). An even more serious issue is that neutron capture by Gd



induces substantial nuclear heating in monazite during irradiation, which may be sufficient to
melt the grains and would certainly anneal any fission tracks produced.
These factors have also ruled out conventional annealing studies dependent on neutron-
induced $^{235}$U fission tracks to assess the geological stability of fission tracks in this mineral.
Alternative thermal annealing experiments have been developed using implanted heavy ion
tracks (e.g. Weise et al., 2009; Ure, 2010), in place of $^{235}$U induced fission tracks. These
methods, in combination with the use of Laser Ablation ICP Mass Spectrometry (LA-ICPMS)
or Electron Probe Microanalysis (EPMA) for determining U concentrations on individual
grains, provide alternatives to the traditional neutron-irradiation approach, thus allowing the
potential of monazite fission track dating to be assessed.

The first published study of fission track dating in monazite was by Shukoljukov and Komarov
(1970), who reported very young ages for two monazite samples from Kazakhstan. The
unexpectedly young results obtained were the first to suggest that fission tracks in monazite
anneal at relatively low temperatures (Shukoljukov and Komarov, 1970). Since this study, the
majority of reported monazite fission track studies have been in conference abstracts (e.g.
Fayon, 2011, Gleadow et al., 2004, and Shipley and Fayon, 2006). Gleadow et al. (2004)
reported preliminary results on several monazite samples revealing fission track ages
considerably younger than corresponding apatite fission track ages, further suggesting that
monazite fission tracks anneal at lower temperatures. This finding was later confirmed by
Shipley and Fayon (2006), who also suggested that annealing rates may vary as a function of
uranium concentration.

A comprehensive annealing study using 300MeV $^{86}$Kr heavy ion tracks in monazite was
published by Weise et al. (2009). Three isochronal annealing sequences were carried out over
schedules of 1, 20 and 100 hr/s on crystals cut parallel to the (100) face. Adapting simplified
apatite annealing models and extrapolating the results to geological timescales they
estimated a closure temperature that "is in all likelihood <50°C and perhaps not much above
ambient".

Ure (2010) carried out further thermal annealing experiments on monazite based on
implanted $^{252}$Cf fission tracks. These were carried out on grains mounted parallel and

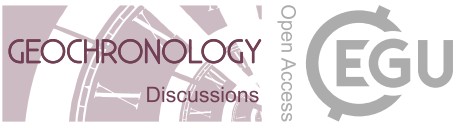

perpendicular to the crystallographic c-axis, with each orientation annealed for 20 minutes
and 1 hour at various temperatures. The results showed that on these short laboratory time
scales, $^{252}$Cf tracks in monazite annealed at lower temperatures when compared to parallel
experiments on Durango Apatite. Further, it was concluded that monazite exhibits similar
anisotropic annealing properties to apatite in that tracks anneal faster perpendicular to the
c-axis compared to the c-axis parallel direction. All of these studies have suggested that fission
tracks in monazite have significant potential as a new ultra-low temperature
thermochronometer, but that further work is required to quantify the annealing kinetics.

Several studies have used heavy ion tracks as proxies for fission track annealing studies in
other minerals. Green et al. (1986) annealed 220-MeV Ni ion tracks in apatite to further
confirm that gaps in the etchability of highly annealed tracks delay the progress of the etchant
along the track length. Sandhu et al., (1990) implanted heavy ion tracks of various energies
(1.67 GeV Nb, 3.54 GeV Pb and 2.38 GeV U) in mica, apatite and zircon, and concluded that
the activation energies for annealing the different energy ion tracks were identical in the
same mineral. Furthermore, they found that in the same mineral, the activation energies for
annealing of tracks formed by $^{252}$Cf fission fragments were also identical to those from the
heavy ion tracks. These studies have shown that the minimum energy required to initiate
annealing is largely independent of the nature and energy of the ion source and rather is a
property of the detector mineral (Sandhu et al., 1990). Because the mass and energy
distributions of both light and heavy fission fragments from $^{252}$Cf are similar to those
produced by spontaneous fission of $^{238}$U, the annealing properties of fission tracks from either
source in monazite should be similar (Fleischer et al., 1975).

In this study, implanted $^{252}$Cf fission tracks are used to constrain the thermal annealing
properties of monazite using a modified etching protocol (Jones et al., 2019). The new
annealing experiments cover a wider time-temperature range than previously reported.
Three alternative kinetic models are then developed that describe the reduction of fission
track lengths as functions of time and temperature. Extrapolation of these models then allows
estimates to be made of the temperature range over which fission-track annealing occurs on
geological timescales.



***2. Experimental methods***
Monazite crystals used in the thermal annealing experiments were separated from the Late
Devonian Harcourt Granodiorite (Victoria, Australia). This is a high-K, calc-alkaline granite
dated by zircon U-Pb and $^{40}$Ar/$^{39}$Ar geochronology to ~370 Ma (Clemens, 2018). Euhedral
monazite crystals range from ~100 – 250 μm in length and are classified as Ce dominant (see
Table 1).

**Table 1.** Average electron microprobe analyses of Harcourt Granodiorite monazite grains

| Element | Mean Wt.% |
|---|---|
| $SiO_2$ | 1.63 ± 0.04 |
| $P_2O_5$ | 27.37 ± 0.15 |
| CaO | 0.45 ± 0.02 |
| $Y_2O_3$ | 2.39 ± 0.05 |
| $La_2O_3$ | 14.13 ± 0.17 |
| $Ce_2O_3$ | 28.54 ± 0.26 |
| $Pr_2O_3$ | 4.45 ± 0.11 |
| $Nd_2O_3$ | 10.61 ± 0.13 |
| $Sm_2O_3$ | 1.80 ± 0.08 |
| $Gd_2O_3$ | 1.34 ± 0.08 |
| $ThO_2$ | 6.31 ± 0.11 |
| $UO_2$ | 0.50 ± 0.04 |
| **Sum Ox%** | **99.52** |

Measurements (± 2σ error) on 81 grains made with a Cameca SX50 electron microprobe using a 10 μm beam width, 50 KeV beam current, 25 KV accelerating voltage and take off angle of 40°.



$^{252}$Cf fission track implantation, measurements and equivalent confined fission track
calculations in this study essentially followed the procedure of Ure (2010). Fifty-five monazite
crystals per sample were attached to double-sided tape on a Teflon block. Then using
tweezers under a stereoscopic microscope, grains were carefully oriented parallel (//) and
perpendicular (⊥) to the crystallographic c-axis (Figure 1), followed by mounting in cold
setting *Struers* Epofix epoxy. For each annealing experiment, two sample mounts were made,
one with grains orientated parallel and another perpendicular to the c-axis. Each sample
mount was then pre-ground using a *Struers* MD-Piano 1200 grinding disc and final polishing
with 6, 3, 1 and 0.25 μm diamond pastes. Polished grain mounts were then exposed to
collimated fission fragments approximately 2 cm from a thin 4mm diameter $^{252}$Cf source
under vacuum for 7 hours to implant a density of ~5 x 10$^6$ tracks/cm$^2$. Tracks were implanted
at an angle of approximately 30° to the polished surface which had been shown to be optimal
for measurement in previous experiments (Ure, 2010).

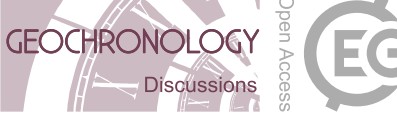


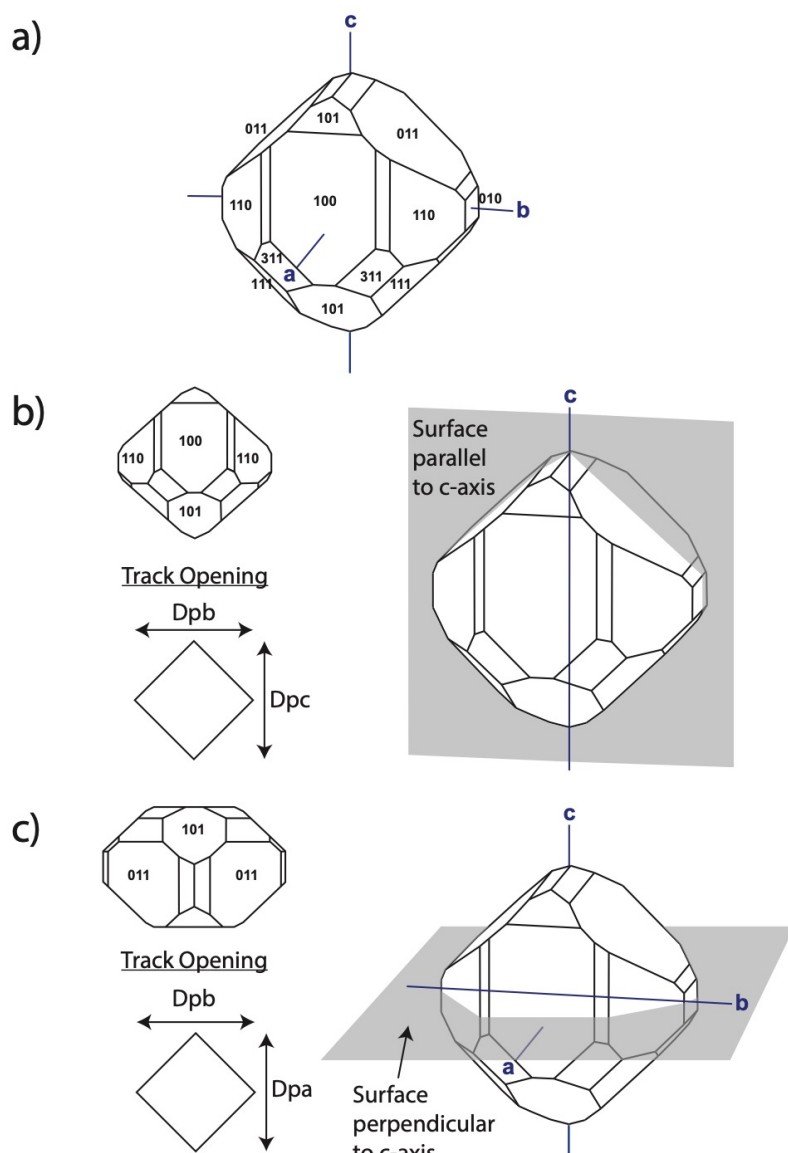


**Figure 1.** (a) Typical monazite crystal with Miller Indices and crystallographic axes. (b) Crystal plane for tracks
implanted on surfaces parallel to the crystallographic c-axis. The shape of the track opening on the etched
surface is a rhombus. Dpb represents diameter of etch pit parallel to b-axis and Dpc is defined as the diameter
of etch pit parallel to c-axis, equivalent to the parameters Dper and Dpar respectively in uniaxial minerals such
as apatite. (c) Crystal plane for tracks implanted perpendicular to c-axis. Track etch pits also tend to be diamond
in shape. Dpa represents diameter of track opening parallel to a-axis. Models from Mindat.org.

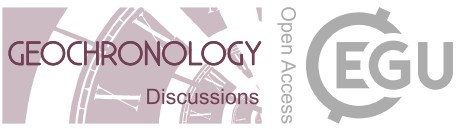




Following track implantation, grains were removed from the mount by dissolving the epoxy
mount in commercial paint-stripper. The loose grains were then annealed in aluminum tubes
in a *Ratek Digital* Dry Block Heater over 1-, 10-, 100- and 1000-hour schedules at
temperatures between 30°C - 400°C. The block heater was covered by a ceramic foam block
for insulation through which a probe could be inserted to monitor temperatures.
Temperature uncertainty is estimated to be ± 2°C. Once each annealing experiment was
completed, the grains were removed from the block heater and re-mounted, polished face
down, on double-sided tape before re-embedding in cold setting *Epofix* epoxy. Etching of each
sample mount was then performed using 6M HCl for 75 minutes at 90°C (Jones et al., 2019).
An example of well-etched $^{252}$Cf fission tracks in this monazite is shown in Figure 2.

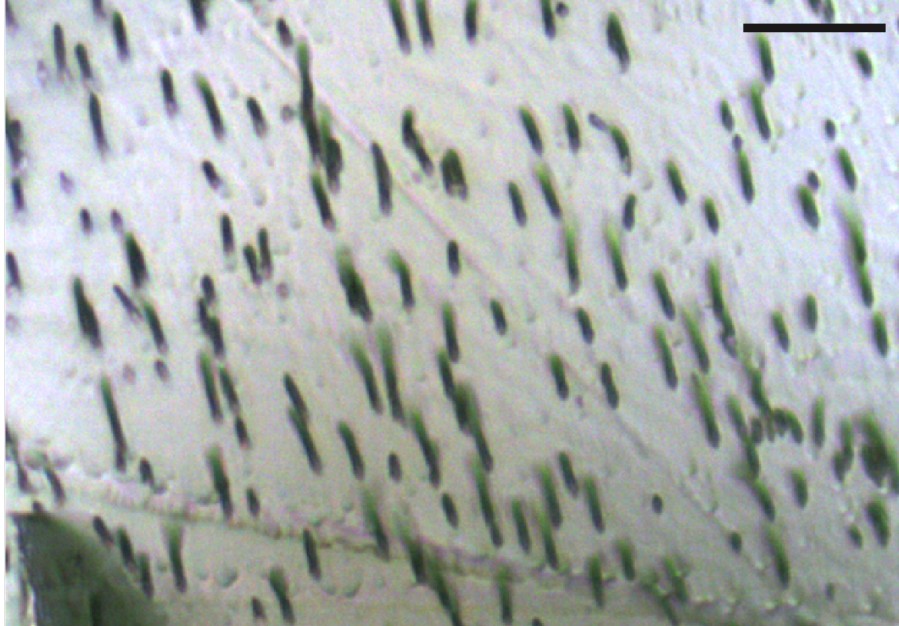


**Figure 2.** Implanted and well-etched $^{252}$Cf fission tracks in Harcourt Granodiorite monazite. Enlarged image taken
with a 100x dry objective, scale bar is 10 μm.

Digital images of all monazite grains in each mount were captured in reflected and
transmitted light using a 100x dry objective on a *Zeiss* Axio Imager M1m motorized
microscope fitted with a *PI* piezo-motor scanning stage and an *IDS* μEye 4 Megapixel USB 3



CMOS digital camera. This was interfaced to a control PC using *Trackworks* software (Gleadow
et al., 2009; 2019). The true 3D lengths of the etched $^{252}$Cf semi-tracks were then measured
from the captured image stacks on a separate computer using *FastTracks* software (Gleadow
et al., 2009; 2019) until a maximum of 500 tracks per sample mount were attained, thus
totaling 1000 tracks per annealing experiment (500 on the c-axis parallel and 500 on the c-
axis perpendicular surfaces). Track length measurements were made using both reflected and
transmitted light images and typically measured over ~30 grains. The surface reflected light
image was used to manually determine the center of the implanted $^{252}$Cf semi-track etch pit,
and the transmitted light stack for determining the position of the track termination by
scrolling down through the image stack to the last image plane where it appeared clearly in
focus. *FastTracks* automatically calculates true track lengths, correcting the vertical focus
depth for the refractive index of monazite, taken to be 1.794.

The equivalent confined track length (l) was then calculated based on a correction for the
small amount of surface lowering during track etching. This surface lowering during etching
on different planes was estimated from diameters of the track etch pits in different directions.
In uniaxial minerals, such as apatite and zircon, the dimensions of track etch pits are
satisfactorily described by the parameters Dpar and Dper (track diameters parallel and
perpendicular respectively to the c-axis, Donelick et al., 2005). However for monoclinic
minerals, such as monazite, the situation is more complex, and we extend this terminology as
shown in Figure 1 with three track diameter measurements, Dpa (diameter parallel to the a-
axis), Dpb (parallel to b) and Dpc (parallel to c), the latter being equivalent to Dpar in apatite
and zircon. The track etch pits in monazite are rhombic in shape and in practice these three
diameter measurements are very similar to each other, so the differences are not critical
(Table 2).




**Table 2.** Average diameters of implanted [252]Cf fission track openings on both parallel and perpendicular surfaces
for each annealing schedule.

| | Dpa (μm) | Dpb (μm) | Dpc (μm) |
|---|---|---|---|
| **Surfaces // c-axis** | | | |
| **1 Hour** | - | 0.62 | 0.61 |
| **10 Hours** | - | 0.64 | 0.60 |
| **100 Hours** | - | 0.62 | 0.63 |
| **1000 Hours** | - | 0.61 | 0.60 |
| **Surfaces ⊥ c-axis** | | | |
| **1 Hour** | 0.62 | 0.61 | - |
| **10 Hours** | 0.62 | 0.63 | - |
| **100 Hours** | 0.63 | 0.64 | - |
| **1000 Hours** | 0.63 | 0.64 | - |
| **Average** | 0.63 | 0.62 | 0.61 |



The track diameter measurements, representing the rate of etching from a point source in
different crystallographic orientations, may be used to estimate the rate of surface lowering
on different surfaces. For (100) surfaces (i.e. parallel to both b- and c-axes), the amount of
surface etching was estimated using measurements of the track width parameter Dpa,
measured on the surface normal to the c-axis (approximately parallel to the a- and b-axes).
Diameter measurements were made for approximately 250 tracks for both surface
orientations in each sample. The amount of surface etching on (100) was approximated by
half the mean Dpa measurement for each sample (Ure, 2010). Knowing the track implantation
angle (30°), allows for the length of the lost portion of the implanted semi-tracks to be
calculated and added to the total track length (Ure, 2010) as illustrated in Figure 3. The
equivalent confined fission track length is then obtained by doubling the corrected mean
semi-track length. For surfaces cut perpendicular to the c-axis (approximately (001)), the
relevant measurement for the surface lowering correction is the half the mean Dpc measured
on the (100) surfaces.




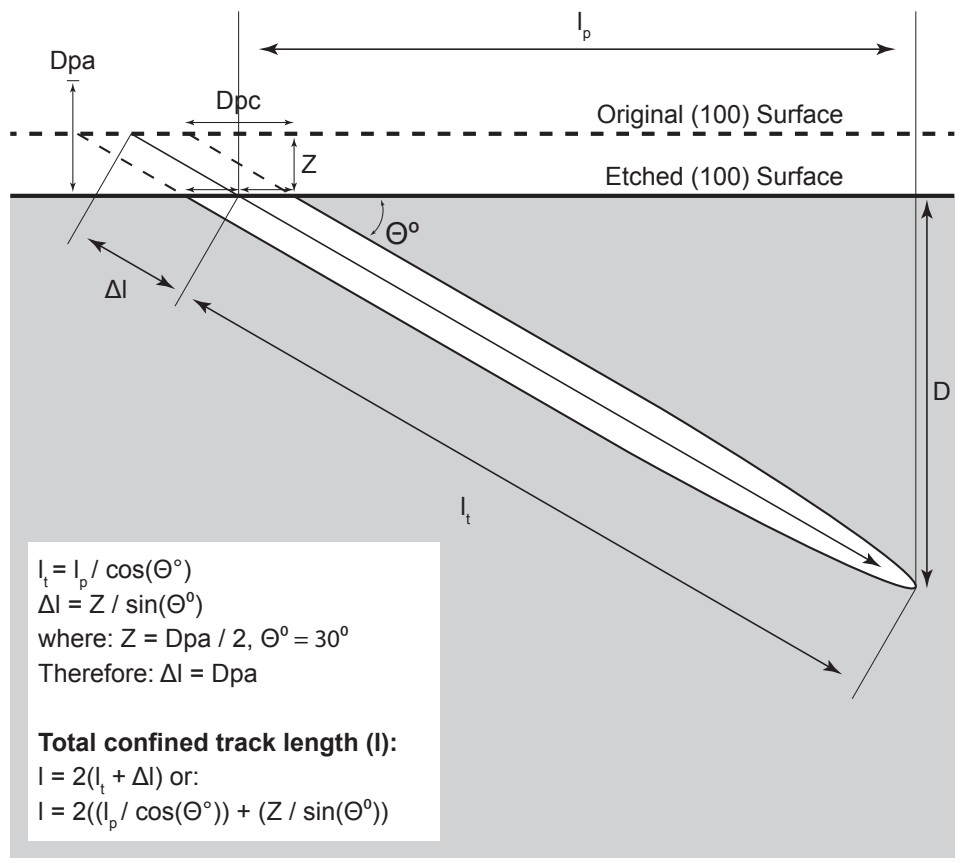

$l_t = l_p / \cos(\Theta°)$
$\Delta l = Z / \sin(\Theta°)$
where: $Z = Dpa / 2$, $\Theta° = 30°$
Therefore: $\Delta l = Dpa$

**Total confined track length (l):**
$l = 2(l_t + \Delta l)$ or:
$l = 2((l_p / \cos(\Theta°)) + (Z / \sin(\Theta°)))$


**Figure 3.** Illustration of the measurements and calculations required to correct semi-track lengths for surface
etching on a (100) face (ie parallel to b and c). Bulk etching removes the original surface by approximately half
the width of the etch pit diameter parallel to the a-axis (Dpa) measured on the ~(001) face (modified from Ure

227 2010).


**_3. Results_**
Table 3 and Figure 4 present the track length measurements from the isochronal annealing
experiments in Harcourt Granodiorite monazite. All length measurements are presented as
mean lengths of equivalent confined fission tracks calculated according to the geometry in
Figure 3 and duplicated on surfaces orientated parallel and perpendicular to the
crystallographic c-axis. The annealing schedules are presented as 1, 10, 100 and 1000 hours
between temperatures of 30°C - 400°C.





Unannealed fission track lengths for all control samples range from 10.12 ± 0.06 − 11.23 ±
0.08 μm, averaging 10.60 ± 0.19 μm. These vary by considerably more than the analytical
uncertainty and possible reasons for this are considered below.  Across all annealing
experiments, mean lengths become progressively shorter, down to a minimum measured
length of 4.88 μm (10 hours, 300°C, perpendicular c-axis). Note that for all the annealed
samples tracks etched on surfaces perpendicular to the crystallographic c-axis are always
shorter than the average length of tracks orientated on surfaces parallel to c. However, the
same is not true for all of the control measurements.

Track length reduction normalized to the mean length for the unannealed control samples
(10.60 μm) are also presented in Table 3. Normalized lengths start at 1 (control sample),
reducing to ~0.5 before dropping abruptly to zero by the next heating step. The shortest mean
track lengths were seen in the 10-hour experiments, where $l/l_0$ decreased to values of 0.502
and 0.460 (300°C, parallel and perpendicular surfaces, respectively).

**Table 3.** Isochronal laboratory annealing data for $^{252}$Cf tracks in the Harcourt Granodiorite monazite (1σ errors).





| Annealing Time | Annealing Temp (℃) | Surface Orientation | ²⁵²Cf Track Length (μm)* | Z (μm) | Calculated Track Length (μm)** | $I/I_0$ (r) | No. of Tracks |
|---|---|---|---|---|---|---|---|
| Control | ~20 | // c-axis | 4.60 ± 0.84 | 0.31 | 10.42 ± 0.08 | 1 | 500 |
| 1 Hour | 50 | // c-axis | 4.29 ± 0.82 | 0.30 | 9.78 ± 0.07 | 0.923 ± 0.010 | 500 |
| 1 Hour | 100 | // c-axis | 4.05 ± 0.69 | 0.32 | 9.36 ± 0.06 | 0.883 ± 0.009 | 500 |
| 1 Hour | 200 | // c-axis | 3.34 ± 0.73 | 0.34 | 8.02 ± 0.07 | 0.757 ± 0.009 | 500 |
| 1 Hour | 300 | // c-axis | 2.90 ± 0.73 | 0.31 | 7.02 ± 0.06 | 0.662 ± 0.008 | 500 |
| 1 Hour | 320 | // c-axis | 2.60 ± 0.82 | 0.31 | 6.42 ± 0.07 | 0.606 ± 0.008 | 500 |
| 1 Hour | 400 | // c-axis | 0 | 0 | 0 | 0 | 0 |
| | | | | | | | |
| Control | ~20 | ⊥ c-axis | 5.00 ± 0.88 | 0.31 | 11.23 ± 0.08 | 1 | 500 |
| 1 Hour | 50 | ⊥ c-axis | 4.27 ± 0.82 | 0.30 | 9.74 ± 0.07 | 0.919 ± 0.009 | 500 |
| 1 Hour | 100 | ⊥ c-axis | 4.01 ± 0.72 | 0.31 | 9.24 ± 0.06 | 0.872 ± 0.008 | 500 |
| 1 Hour | 200 | ⊥ c-axis | 3.25 ± 0.70 | 0.32 | 7.76 ± 0.06 | 0.732 ± 0.007 | 500 |
| 1 Hour | 300 | ⊥ c-axis | 2.60 ± 0.74 | 0.32 | 6.48 ± 0.06 | 0.611 ± 0.007 | 500 |
| 1 Hour | 320 | ⊥ c-axis | 2.44 ± 0.73 | 0.33 | 6.18 ± 0.07 | 0.583 ± 0.007 | 500 |
| 1 Hour | 400 | ⊥ c-axis | 0 | 0 | 0 | 0 | 0 |
| | | | | | | | |
| Control | ~20 | // c-axis | 4.82 ± 0.57 | 0.32 | 10.90 ± 0.05 | 1 | 500 |
| 10 Hours | 50 | // c-axis | 4.20 ± 0.71 | 0.30 | 9.60 ± 0.06 | 0.906 ± 0.007 | 500 |
| 10 Hours | 100 | // c-axis | 3.82 ± 0.62 | 0.33 | 8.94 ± 0.06 | 0.843 ± 0.007 | 500 |
| 10 Hours | 150 | // c-axis | 3.43 ± 0.64 | 0.34 | 8.22 ± 0.06 | 0.775 ± 0.007 | 500 |
| 10 Hours | 200 | // c-axis | 3.17 ± 0.60 | 0.30 | 7.54 ± 0.06 | 0.711 ± 0.006 | 500 |
| 10 Hours | 250 | // c-axis | 2.77 ± 0.69 | 0.34 | 6.88 ± 0.06 | 0.649 ± 0.006 | 500 |
| 10 Hours | 300 | // c-axis | 2.03 ± 0.72 | 0.32 | 5.32 ± 0.06 | 0.502 ± 0.006 | 500 |
| 10 Hours | 350 | // c-axis | 0 | 0 | 0 | 0 | 0 |
| | | | | | | | |
| Control | ~20 | ⊥ c-axis | 4.65 ± 0.53 | 0.33 | 10.62 ± 0.05 | 1 | 500 |
| 10 Hours | 50 | ⊥ c-axis | 4.15 ± 0.69 | 0.31 | 9.54 ± 0.06 | 0.900 ± 0.007 | 500 |
| 10 Hours | 100 | ⊥ c-axis | 3.81 ± 0.54 | 0.30 | 8.82 ± 0.05 | 0.832 ± 0.006 | 500 |
| 10 Hours | 150 | ⊥ c-axis | 3.40 ± 0.68 | 0.30 | 8.00 ± 0.06 | 0.755 ± 0.007 | 500 |
| 10 Hours | 200 | ⊥ c-axis | 3.09 ± 0.66 | 0.30 | 7.38 ± 0.06 | 0.696 ± 0.007 | 500 |
| 10 Hours | 250 | ⊥ c-axis | 2.63 ± 0.66 | 0.33 | 6.56 ± 0.06 | 0.619 ± 0.006 | 500 |
| 10 Hours | 300 | ⊥ c-axis | 1.81 ± 0.71 | 0.32 | 4.88 ± 0.06 | 0.460 ± 0.006 | 500 |
| 10 Hours | 350 | ⊥ c-axis | 0 | 0 | 0 | 0 | 0 |
| | | | | | | | |
| Control | ~20 | // c-axis | 4.85 ± 0.75 | 0.30 | 10.90 ± 0.07 | 1 | 500 |
| 100 Hours | 30 | // c-axis | 4.46 ± 0.90 | 0.30 | 10.12 ± 0.08 | 0.955 ± 0.009 | 500 |
| 100 Hours | 50 | // c-axis | 4.19 ± 0.94 | 0.31 | 9.62 ± 0.08 | 0.908 ± 0.009 | 500 |
| 100 Hours | 100 | // c-axis | 3.75 ± 0.68 | 0.30 | 8.70 ± 0.06 | 0.821 ± 0.008 | 500 |
| 100 Hours | 150 | // c-axis | 3.32 ± 0.80 | 0.34 | 7.98 ± 0.07 | 0.753 ± 0.008 | 500 |
| 100 Hours | 200 | // c-axis | 3.04 ± 0.70 | 0.34 | 7.44 ± 0.06 | 0.702 ± 0.007 | 500 |
| 100 Hours | 250 | // c-axis | 2.51 ± 0.73 | 0.32 | 6.28 ± 0.07 | 0.592 ± 0.007 | 500 |
| 100 Hours | 300 | // c-axis | 0 | 0 | 0 | 0 | 0 |
| 100 Hours | 350 | // c-axis | 0 | 0 | 0 | 0 | 0 |
| | | | | | | | |
| Control | ~20 | ⊥ c-axis | 4.50 ± 0.76 | 0.30 | 10.20 ± 0.07 | 1 | 500 |
| 100 Hours | 30 | ⊥ c-axis | 4.26 ± 0.84 | 0.32 | 9.80 ± 0.08 | 0.925 ± 0.010 | 500 |
| 100 Hours | 50 | ⊥ c-axis | 4.05 ± 0.83 | 0.33 | 9.42 ± 0.07 | 0.889 ± 0.009 | 500 |
| 100 Hours | 100 | ⊥ c-axis | 3.65 ± 0.63 | 0.31 | 8.54 ± 0.06 | 0.806 ± 0.008 | 500 |
| 100 Hours | 150 | ⊥ c-axis | 3.31 ± 0.74 | 0.32 | 7.90 ± 0.07 | 0.745 ± 0.008 | 500 |
| 100 Hours | 200 | ⊥ c-axis | 3.01 ± 0.69 | 0.32 | 7.28 ± 0.06 | 0.687 ± 0.008 | 499 |
| 100 Hours | 250 | ⊥ c-axis | 2.49 ± 0.53 | 0.32 | 6.24 ± 0.05 | 0.589 ± 0.006 | 500 |
| 100 Hours | 300 | ⊥ c-axis | 0 | 0 | 0 | 0 | 0 |
| 100 Hours | 350 | ⊥ c-axis | 0 | 0 | 0 | 0 | 0 |
| | | | | | | | |
| Control | ~20 | // c-axis | 4.46 ± 0.64 | 0.30 | 10.12 ± 0.06 | 1 | 500 |
| 1000 Hours | 50 | // c-axis | 4.03 ± 0.60 | 0.30 | 9.26 ± 0.06 | 0.874 ± 0.008 | 500 |
| 1000 Hours | 150 | // c-axis | 3.18 ± 0.54 | 0.31 | 7.60 ± 0.05 | 0.717 ± 0.007 | 500 |
| 1000 Hours | 200 | // c-axis | 3.04 ± 0.74 | 0.30 | 7.28 ± 0.07 | 0.687 ± 0.007 | 500 |
| 1000 Hours | 250 | // c-axis | 2.60 ± 0.96 | 0.31 | 6.42 ± 0.09 | 0.606 ± 0.007 | 500 |
| 1000 Hours | 275 | // c-axis | 0 | 0 | 0 | 0 | 0 |
| | | | | | | | |
| Control | ~20 | ⊥ c-axis | 4.58 ± 0.65 | 0.31 | 10.40 ± 0.06 | 1 | 500 |
| 1000 Hours | 50 | ⊥ c-axis | 3.99 ± 0.60 | 0.30 | 9.18 ± 0.06 | 0.866 ± 0.008 | 500 |
| 1000 Hours | 150 | ⊥ c-axis | 3.15 ± 0.52 | 0.31 | 7.56 ± 0.05 | 0.713 ± 0.006 | 500 |
| 1000 Hours | 200 | ⊥ c-axis | 2.79 ± 0.59 | 0.33 | 6.88 ± 0.05 | 0.649 ± 0.006 | 500 |
| 1000 Hours | 250 | ⊥ c-axis | 2.02 ± 1.08 | 0.33 | 5.36 ± 0.16 | 0.506 ± 0.008 | 187 |
| 1000 Hours | 275 | ⊥ c-axis | 0 | 0 | 0 | 0 | 0 |

\* ± sd, \*\* ± se

Z is the amount of surface lowering due to bulk etching

$I/I_0$ (r) has been normalized to average control sample of 10.60 μm







### *4. Discussion*

The average track length for the unannealed control samples across all analyses is 10.60 $\pm$
0.19 μm which is slightly shorter but within error of the 11.30 $\pm$ 0.36 μm mean length reported
by Ure (2010) for a smaller number of tracks in a different monazite of unknown composition.
Weise et al. (2009) calculated a mean range 8.30 $\pm$ 0.62 μm for a heavy fission fragment and
10.80 $\pm$ 0.52 μm for a light fission fragment for $^{235}$U fission in monazite. This combines to give
a total latent track length of ~19 μm. However, it has long been known (e.g. Fleischer et al.,
1975) that the lengths of etched fission tracks are significantly shorter than the total range of
the fission fragments due to a 'length deficit' of unetchable radiation damage towards the
end of the track. Weise et al. (2009) calculated the length deficit for a unannealed confined
fission track in monazite to be 6-7 μm, making the etchable length for induced $^{235}$U fission
tracks ~12-13 μm. Our measurements for the unannealed control samples are on average ~1-
2 μm shorter than these estimates, suggesting that the length deficit may be closer to 8μm
(~4μm at each end) at least for the $^{252}$Cf tracks used here.  The mean track lengths reported
here are also broadly consistent with measured lengths of spontaneous $^{238}$U confined tracks,
reported to be ~10 μm (Weise et al., 2009).

There is a difference of 1.11 μm between the longest and shortest mean track lengths in
control samples across the experiments. This is substantial and significantly greater than the
measurement uncertainty. It is known that newly produced fission tracks in apatite undergo
rapid annealing at ambient temperatures (Donelick et al., 1990) from the moment the track
is formed in the crystal lattice until the track is etched.  It was not clear whether this was due
to short-term thermal annealing or some non-thermal annealing mechanism. Belton (2006)
and Tamer and Ketcham (2020) also found similar effects in a series of ambient temperature
annealing experiments on freshly induced $^{235}$U fission tracks in various apatites. The results
showed the tracks reduced in length by 0.32 - 0.70 μm between 39 seconds and 1.88 days
after irradiation and continued to shorten measurably over decades. While the exact amount
of time between $^{252}$Cf track implantation and etching for each individual control sample was
not recorded in this study, the considerable length differences in the control samples suggest



that ambient temperature annealing may also occur in monazite, and probably to an even
greater degree than in apatite.

Differing degrees of ambient temperature annealing may also be the reason why mean track
lengths in monazite control samples cut perpendicular to the c-axis were not always shorter
than in those parallel to the c-axis, as was invariably the case for all experiments at higher
temperatures. Further, Figure 4 shows that for all the isochronal experiments, the annealing
curves exhibit an initial length reduction of ~5-10% before the 50°C annealing step, a feature
not observed in annealing experiments in other minerals. This may be due to the mean track
length for the control samples not having reached a stable value at ambient temperature
prior to the thermal annealing experiments.

Importantly, over the temperature range studied, no conditions have been identified where
the tracks are totally stable (Figure 4), even for experiments conducted at 30°C. Figure 2 also
shows that there is a gradual reduction in $l/l_0$ with temperature, followed by accelerated
reduction from ~0.580 to zero. For this reason, values of $l/l_0$ < ~0.5 are rarely encountered,
with only two slightly lower values (0.460 and 0.488) being observed amongst all 52
experiments.  This is a similar behaviour to that seen in apatite and zircon (e.g. Green et al.,
1986; Yamada et al., 1995). Relatively less difference was observed between the averaged
track length reduction of the 100- and 1000- hour schedules compared to the shorter
annealing times.


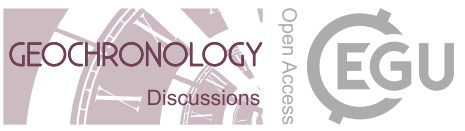

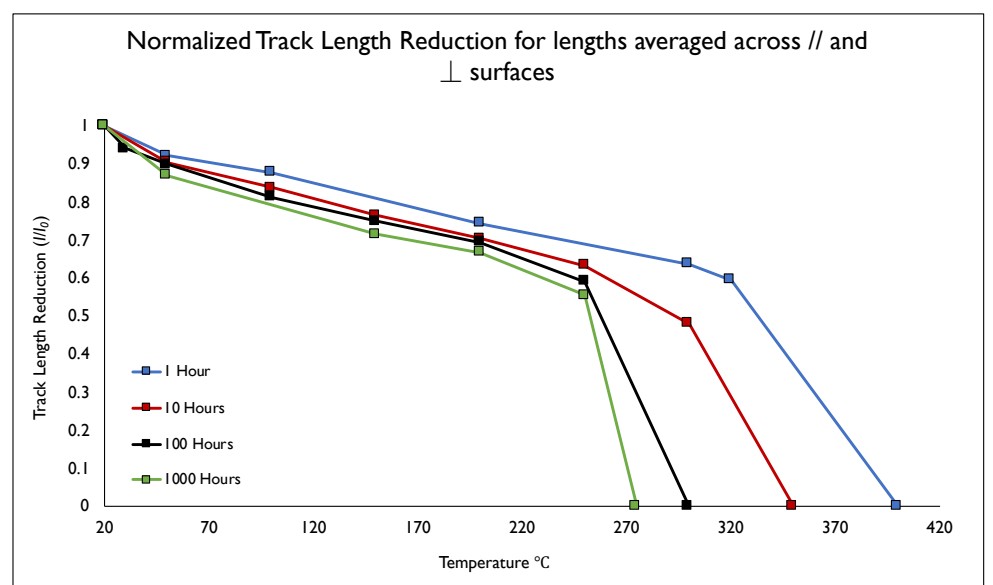

**Figure 4.** Track length reduction ($l/l_0$) against temperature for calculated equivalent confined fission tracks in Harcourt Granodiorite monazite. The track length reduction values are averaged across both parallel (//) and perpendicular ($\perp$) surfaces with the normalized track length ($l/l_0$) values being calculated from the average length of the unannealed control samples (10.60 $\mu$m).

In all annealed samples, the mean equivalent confined track length was always less than that for the unannealed control samples. As annealing progresses, the mean track lengths are reduced, and become consistently anisotropic with crystallographic orientation, although the differences are small and all within errors. Tracks implanted at 30° dip to polished surfaces oriented perpendicular to the crystallographic c-axis always have shorter mean track lengths than those at 30° to the c-axis parallel surfaces. On both these surface orientations the dips were constant but there was limited control on the azimuth orientations of the collimated tracks, so the exact relationship to crystallographic orientation is not clear. However, the distribution of track orientations will cover a different range on the two surfaces so that anisotropy of annealing can clearly be detected. As annealing progresses, the amount of anisotropy generally increases across all annealing schedules for the two surface orientations with the exception of 100 hours. That is, tracks on surfaces orientated perpendicular to the crystallographic c-axis anneal faster with increasing temperature. Anisotropy is still present in the 100-hour schedule, but no clear increase in the difference between calculated confined track lengths is apparent for the two differently oriented surface planes. Anisotropy is

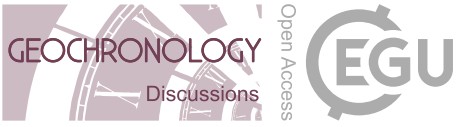



greatest in the 1000 hours, 250°C experiments, where there is a ~1.06 μm difference between
the two surface orientations (Figure 5). This is possibly due to only 187 semi-track lengths
being measured in the c-axis perpendicular aliquot (as most were completely annealed)
compared to 500 in the parallel aliquot.

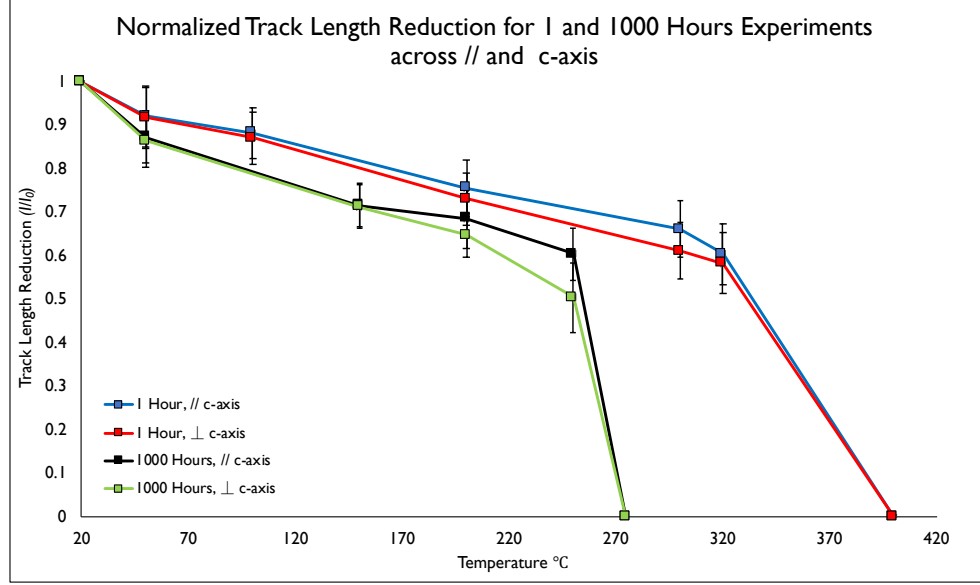


**Figure 5.** Track length reduction ($l/l_0$) against temperature for calculated equivalent confined fission tracks for 1
and 1000 hour experiments for both surface orientations. The normalized track length ($l/l_0$) values are calculated
from the average length of the control samples (10.60 μm). Error bars refers to 1σ errors.

Figure 6 shows the relationship between the standard deviation and mean track length for
the length distributions of single fission fragment [252]Cf tracks. The results vary between 0.52
and 1.08 μm and are mostly consistent with a mean of 0.71 μm but with considerable scatter.
The results suggest an increase in standard deviation at short mean lengths, as is observed
for confined track length measurements in apatite during annealing (e.g. Green et al., 1986,
Fig 3) because of increasing anisotropy. For monazite, the amount of anisotropy also appears
to increase as the mean track length decreases giving an increase in dispersion of individual
track lengths, and hence standard deviation. The most extreme annealing observed is for the
1000 Hours, 250°C experiment, with a standard deviation of 1.08 μm, which shows the
greatest degree of anisotropy.






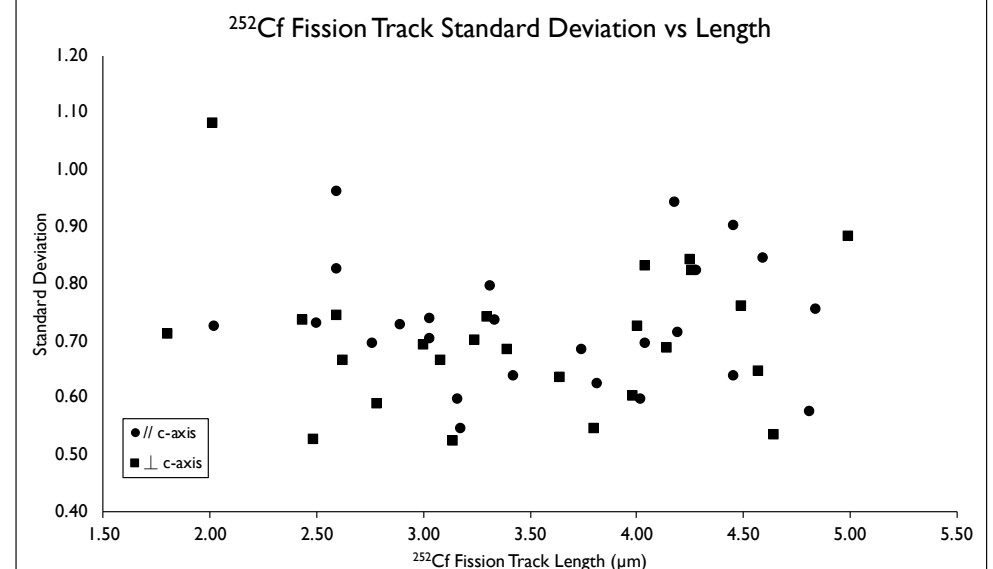


**Figure 6.** Standard deviation of ²⁵²Cf fission-track length distributions plotted against their average track lengths

for both parallel and perpendicular surfaces across all experiments.

### *5. The Arrhenius Plot*

Results of the Harcourt Granodiorite monazite annealing experiments are shown on an

Arrhenius plot of log time versus inverse absolute temperature in Figure 7. Results are

averaged across both surface orientations, and the normalized track length (r=$l$/$l_0$) values are

calculated relative to the average length of the unannealed control samples ($l_0$ = 10.60 μm).

In the plot, normalized track length values in a particular range are represented by the same

symbol and exhibit linear trends with positive correlation. To extrapolate laboratory

annealing results in Arrhenius plots to geological timescales, three types of model fitting have

traditionally been used to determine a functional form of the fission track annealing kinetics,

i.e. the 'parallel model' and two variations of the 'fanning model' (Laslett et al., 1987).

363



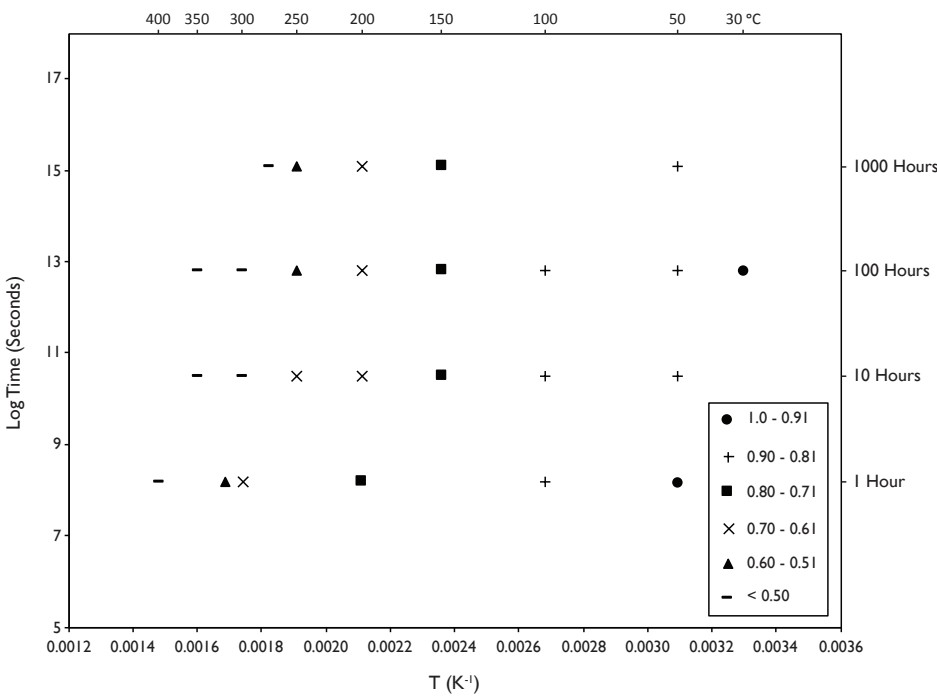

364

**Figure 7.** Arrhenius plot of experimental data using calculated equivalent confined fission track lengths in
Harcourt Granodiorite monazite. Each point represents two annealing experiments that have been averaged
across both orientations (parallel and perpendicular to c-axis). Different degrees of track length reduction (*r*) are
shown by different symbols. Inverse absolute temperature in Kelvins shown on the x-axis and corresponding
temperatures in °C along the top.

### 5.1 Parallel Linear Model

As a starting point, the annealing data of this study will be tested with the 'parallel model'
that has straight line contours (Laslett et al., 1987):

$$\ln(t) = A\,(r) + B\,/\,T \tag{1}$$

Where $t$ = annealing time; $T$ = annealing temperature (K); $A\,(r)$ = intercept of the lines (at $1/T$
= 0), which is a function of the most reliable values of normalized mean length $r$; and $B$ is the
slope, which is a constant for all degrees of annealing. The intercept $A\,(r)$ is subject to the
following constraints: (1) $A\,(r)$ decreases monotonically with increasing $r$; and (2) $A\,(r = 1) \rightarrow$



- ∞ when $t \rightarrow 0$, $T \rightarrow 0$. It should be noted that $r = 0$ for finite values of $t$ and $T$ provided they
are large enough, in practice.

383          The fully parameterized parallel model has the form:


$r = c_1 + c_2\, A(r) + \varepsilon$

$= c_1 + c_2\, [\ln(t) - B/T] + \varepsilon$                                                   (2)

or

$g(r;\, a,\, b) = C_0 + C_1 \ln(t) + C_2/T + \varepsilon$                                    (3)


Where $C_0 = c_1$; $C_1 = c_2$; $C_2 = -c_2 B$; $g(r;\, a,\, b)$ is a transform containing $r$ and two parameters $a$
and $b$; and $\varepsilon$ represents errors or residuals. $\varepsilon$ is assumed to be normally distributed with mean
$\mu = 0$ and constant variance $\sigma^2$. This assumption can be checked by residual plot for the model
in Figure 9. A single Box-Cox transformation was adopted and was found to be better suited
to the data than the double Box-Cox (Box and Cox, 1964):

$g(r;\, a,\, b) = [\, \{\, (1 - r^b)\, /\, b\}^a - 1]/a$                                    (4)


In the model of Eq. 3, parameters and uncertainties (standard error) were evaluated for the
data sets in Table 4 as follows:

$a = 1$,                              $b = 3.72$

$C_0 = -0.440275 \pm 0.034626$,          $C_1 = -0.019504 \pm 0.002284$


and

$C_2 = 437.315478 \pm 10.901345$


**5.2 Fanning Linear Model**
The fanning Arrhenius plot of Laslett et al. (1987) has slopes of contour lines that change with
a variation of activation energy E with the degree of annealing. In this case, Eq. 1 becomes:





$\ln(t) = A(r) + B(r) / T$      (5)

where both slope $B(r)$ and intercept $A(r)$ are a function of $r$. A first order assumption of this
equation is that $A(r)$ is a negative linear function of $B(r)$:

$A(r) = c_3 - c_4\, B(r)$      (6)

where $c_3$ and $c_4$ are constants, by analogy with the 'compensation law' for diffusion (e.g., Hart,
1981). This causes the contours to fade and meet at a single point on the Arrhenius plot.
Combining Eqs. 4 and 5 becomes:

$\ln(t) = A\!* + B(r)\,[(1/T) - (1/T_0)]$      (7)

where $A\!* = c_3$; and $1/T_0 = c_4$. $T_0$ is known as the "critical temperature", which is the
temperature of the 'cross-over point' of the fading contours (e.g. Crowley et al., 1991). Solving
Eq. 6 for $B(r)$ gives:

$B(r) = (\ln(t) - A\!*)/[(1/T) - (1/T_0)]$      (8)

Constraints for slope $B(r)$ are: (1) $B(r)$ decreases monotonically with increasing $r$; and (2) $B(r =$
$1) \rightarrow 0$ when $\ln(t) \rightarrow A\!*$, $T \rightarrow 0$. The fully parameterized model is given as:

$r = c_1 + c_2\, B(r) = c_1 + c_2\, [\{\ln(t) - A\!*\} / \{(1/T) - (1/T_0)\}] + \varepsilon$      (9)

or

$r = C_0 + (C_1 \ln(t) + C_2)/[(1/T) - C_3] + \varepsilon$      (10)

where $C_0 = c_1$; $C_1 = c_2$; $C_2 = -c_2 A\!*$; and $C_3 = 1/T_0$.

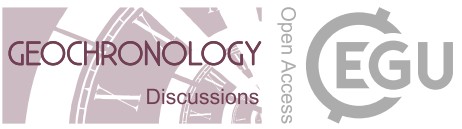

When $C_3 = 0$, this assumes an infinite critical temperature (i.e., $T_0 = \infty$). The equation can be
rearranged to:

$r = C_0 + C_1 \text{ T} \ln t + C_2 T + \varepsilon$                   (11)


The number of parameters is reduced from four to three, simplifying the equation. The
parameters and uncertainties (standard error) for the models in Eq. 10 were calculated as
follows:

$C_0 = 1.374 \pm 0.02698,$        $C_1 = -0.001105 \pm 0.00007301$


and

$C_2 = -0.00002979 \pm 0.000004959$


In the case where $T_0 \neq \infty$, Eq. 9 was adopted for the fitting calculation. The parameters and
uncertainties were evaluated as follows:

$C_0 = 1.227 \pm 0.09638,$        $C_1 = - -0.00002418 \pm 0.000005221,$


$C_2 = -0.0005491 \pm 0.0003005$


and

$C_3 = - 0.0005542 \pm 0.0003468$


Both single and double Box-Cox transforms were applied to Eqs. 10 and 11. A single Box-Cox
transformation was better suited to fit the data; however, it did not statistically improve the
models. A t-test found that Eq. 11 with a single Box-Cox transformation had a $P$-value of 0.096.
Generally, a $P$-value $< 0.05$ suggests strong evidence against the null hypothesis and that it
should be rejected. Whereas a $\rho$-value $> 0.05$ indicates weak evidence against the null
hypothesis, failing to reject it. In the case of Eq. 11 the null hypothesis is the equation without





a transformation and the alternative is to include the single Box-Cox transformation. Using a
similar form of test for Eq. 10 found that the $C_3$ constant produced a $\rho$-value of 0.123. This
high *P*-value suggests that the constant is not preferred and that the model from Eq. 11 is
more parsimonious. For these reasons, the final fanning models are presented with no
transformation (Eq. 10 and 11) and their assumptions can be checked in Figure 9.

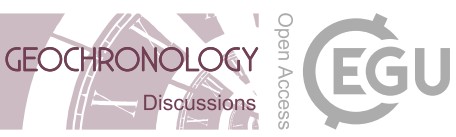



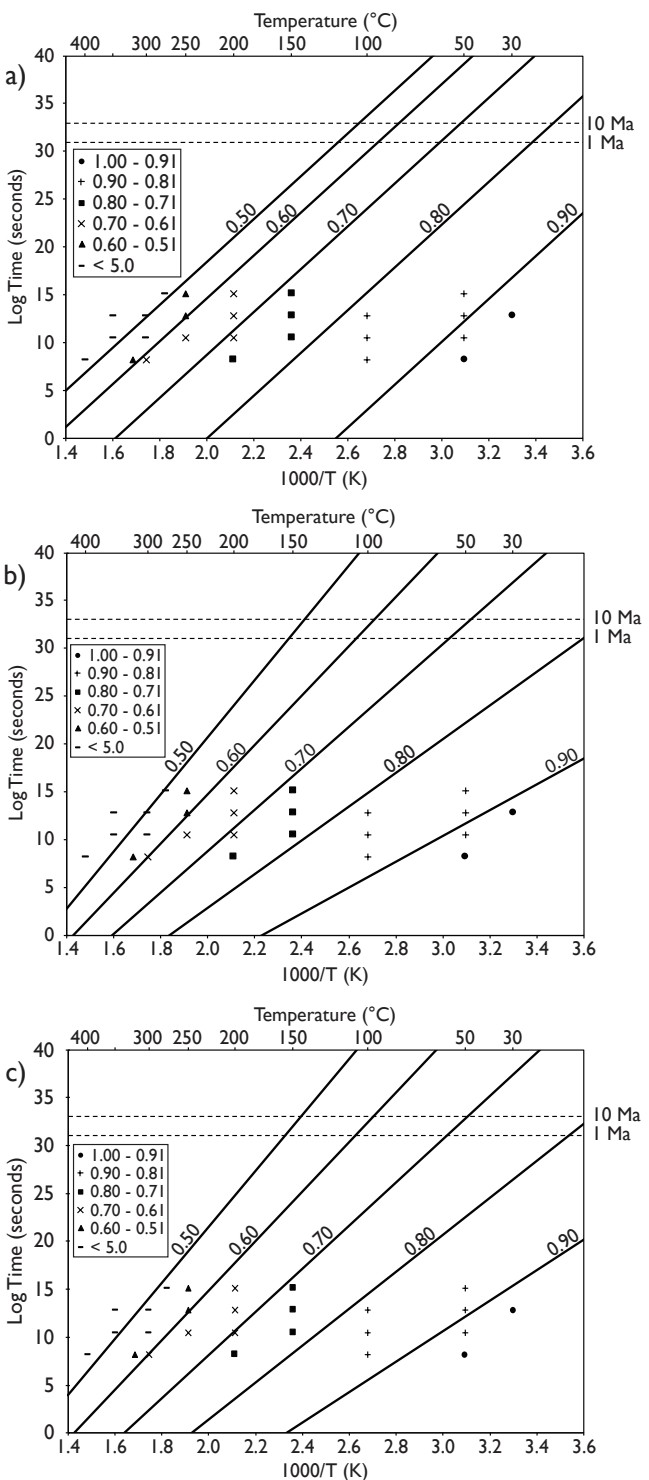






**Figure 8.** Arrhenius plots with fitted lines extrapolated to geological timescales. (a) parallel model; (b) fanning
model ($T_0 \neq \infty$); and (c) fanning model ($T_0 = \infty$). Each plot was obtained by adopting specific equations: (a) Eq. 3;
(b) Eq. 10; and (c) Eq.11 (see text), and parameters as in Table 4. Values of normalized mean length (*r*) for each
contour are indicated on the plots, ranging from 0.90 to 0.50. Symbols are the same as for Figure 5.

**Table 4.** Results of the Arrhenius model fitting calculations including estimated temperatures (°C ± 2σ error) for
the monazite partial annealing zone (MPAZ). Note the $T_0 \neq \infty$ estimated MPAZ has no error listed as it is not
possible to reliably calculate the confidence intervals.

|  | Parallel Model | Fanning Model | |
|---|---|---|---|
|  |  | $T_0 \neq \infty$ | $T_0 = \infty$ |
| **Model Equation** | Eq.3 | Eq. 10 | Eq. 11 |
| **Coefficient of Determination ($R^2$)** | 0.99 | 0.97 | 0.97 |
| **Bottom of MPAZ (2σ) (℃)** |  |  |  |
| **Heating Duration:** |  |  |  |
| **1 Ma** | -39.64 ± 6.14 | -82.52 | -64.30 ± 13.30 |
| **10 Ma** | -44.11 ± 6.49 | -89.54 | -71.12 ± 13.78 |
| **Top of MPAZ (2σ) (℃)** |  |  |  |
| **Heating Duration:** |  |  |  |
| **1 Ma** | 116.47 ± 16.06 | 153.75 | 157.33 ± 20.55 |
| **10 Ma** | 101.48 ± 16.60 | 140.25 | 143.26 ± 21.70 |





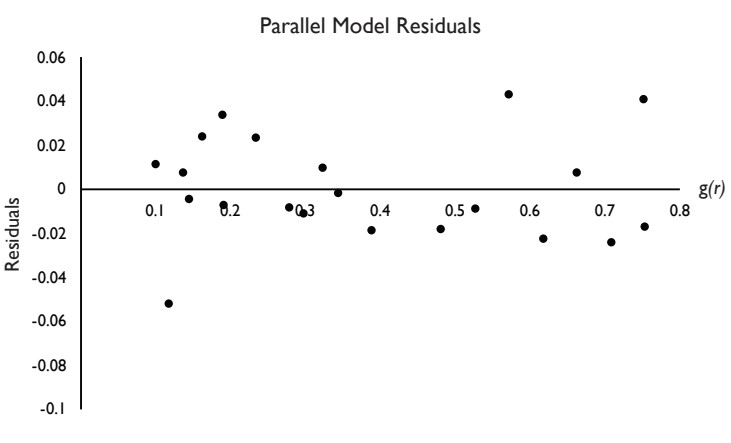

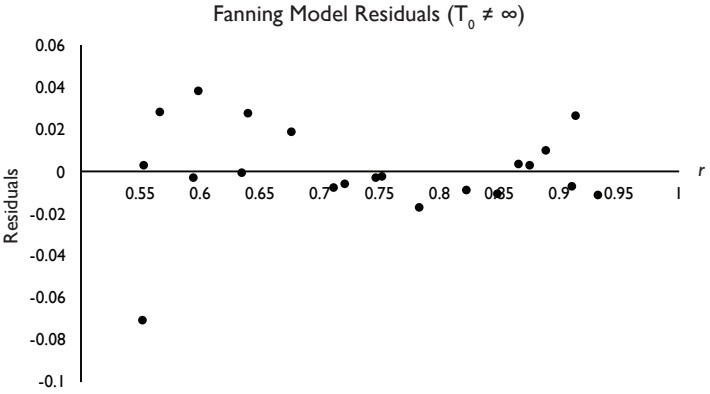

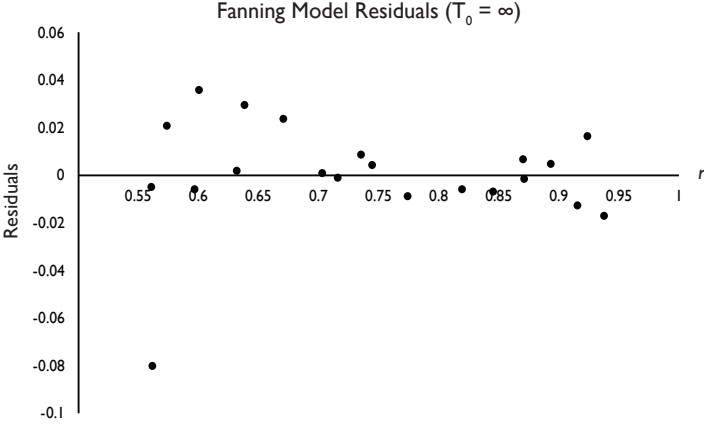


**Figure 9.** Residual Plots for the best fitting calculations for each model ($\varepsilon$ in Eqs. 3, 10 and 11). Each point
represents one annealing experiment.




### 5.3 Comparison of Arrhenius Models

Table 4 and Figure 8 present the results of the model fitting calculations and their associated
Arrhenius plots. The models show the full data set with contours of equal length reduction
extrapolated to geological timescales. The parallel model, which has a constant activation
energy with decreasing $r$, statistically describes the data slightly better than the two fanning
models (coefficient of determination of 0.99 compared to 0.97 for both fanning models).
Nevertheless, the two fanning models, which have an increasing activation energy with
decreasing $r$, still describe the data very well. Although the coefficient of determination of the
two fanning models are equal, the $P$-value of 0.128 for constant $C_3$ in Eq. 10 suggests that the
simpler model is the more favourable. Residual plots for each model (Figure 7) show no clear
structure suggesting that the residuals do not contradict the linear assumption of the models.
In previous studies (e.g. Crowley et al., 1991; Laslett et al., 1987; Yamada et al., 1995), both
fanning models have a Box and Cox (1964) or similar type of transformation on the left-hand
side of the equation, but because they did not statistically improve them, they were
abandoned in this study. The fanning models, as they stand, explain the data very well, and
in general, when constructing empirical models to be used as the basis of prediction, simple
models with less fitted parameters are generally preferable (Laslett et al., 1987). Regardless
of using a transformation or not, all models presented in this study give a statistically
satisfactory description of the available data.

When comparing the models over laboratory timescales, little difference is observed between
them, particularly at length reductions < 0.80. The 0.90 track reduction contour shows the
largest difference over laboratory timescales, where both fanning models splay out to lower
temperatures. This suggests that fission tracks in monazite are even more sensitive to low
temperature annealing in the fanning models compared to the parallel model. As with all such
annealing studies, differences in annealing are magnified when the data are extrapolated to
geological timescales. The assumption underlying such extrapolations is that track annealing
results from the same physical mechanism under both laboratory and geological conditions.
All models show that significant reduction in the etchable lengths of fission tracks takes place
at ambient and lower temperatures (< 20°C) over geological timescales and that monazite is
particularly sensitive to low temperature thermal annealing. Considerably more track





shortening would occur in the shallow upper crust between temperatures of ~50 - 160°C over
geological timescales of 1 – 10 Ma. Complete annealing of fission tracks occurred very rapidly
when the equivalent confined track length reduction decreased below ~0.5.

Weise et al. (2009) presented a linear fanning model that used contours representing the
amount of track length reduction of implanted Kr-tracks in monazite rather than the
normalised reduction ($l/l_0$) as used here. However, similarities can be seen between the
different approaches. Both models show considerable track annealing at ambient surface
temperatures or below over geological timescales. That is, they are in agreement that a total
fission track stability zone is absent for monazite.

**6. Estimation of the monazite partial annealing zone**
Geological temperature ranges for the monazite partial annealing zone (MPAZ) were
calculated by extrapolating model equations to the geological timescale with parameters
derived from the annealing experiments (Table 4). The lower temperature limit of the MPAZ
has been defined as $l/l_0$ = 0.95, since a track length reduction at the 5% level should be clearly
detectable under the microscope. The higher temperature limit of the MPAZ is defined at $l/l_0$
= 0.50, which corresponds with the final rapid fading of tracks observed in this study. The
parallel model (Figure 8a) shows estimates of the MPAZ for a heating duration of $10^7$ years ~-
44 – 101°C. Both fanning models estimate a wider temperature range for the same heating
duration: -89 – 140°C ($T_0 \neq \infty$); and -71 - 143°C ($T_0 = \infty$). The uncertainties of estimated
temperatures are ca. $\pm$ 6 - 21°C for Eqs. 3 and 11 (2 standard errors). The bootstrapping
method for calculating the uncertainties of the estimated MPAZ temperatures could not be
confidently calculated for Eq. 10 and therefore error estimates have not been included for
this model. The inability to confidently calculate the uncertainties of Eq. 10 further supports
the choice of Eq. 11 ($T_0 = \infty$) as the preferred fanning model. Of the two remaining estimates
for the MPAZ range (Eqs. 3 and 11), based on the coefficients of determination, the parallel
model is slightly preferable. However, the fanning model of Eq. 11 also describes the data
almost as well and should not be ruled out. In fact, annealing studies of other minerals such
as zircon and apatite have shown a fanning model to best fit their respective datasets (e.g.
Ketcham et al., 1999; Laslett et al., 1987; Yamada et al., 1995). Taking the fission track closure

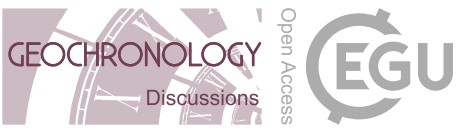

temperature ($T_c$) to be approximately the middle of the MPAZ (Yamada et al., 1995), predicted
closure temperatures for the monazite fission track system range between ~45 - 25°C over
geological timescales of $10^6 – 10^7$ years. The These results are consistent with the findings of
Weise et al. (2009), the only other study to estimate a $T_c$ for the monazite fission track system,
who estimated $T_c$ to be < 50°C and perhaps not much above ambient.

### 7. Conclusions
Using implanted $^{252}$Cf semi-tracks, isochronal annealing experiments were performed on
monazite crystals from the Harcourt Granodiorite in Central Victoria. Semi-track lengths were
measured and combined with an estimate of the degree of surface etching to give calculated
equivalent confined fission track lengths. The unannealed equivalent confined fission track
lengths (control samples) have a mean length of 10.60 ± 0.19 μm, which is broadly consistent
with the measured lengths of spontaneous $^{238}$U confined tracks reported by Weise et al.
(2009). As annealing progresses, the mean calculated confined track length decreases
anisotropically, with tracks on surfaces perpendicular and parallel to the crystallographic c-
axis annealing at measurably different rates.

Using the equations of Laslett et al. (1987), three empirical models describe the data
remarkably well, with the parallel Arrhenius plot fitting the data slightly better than two
alternative fanning models. The differences between models are negligible, however, and, in
line with experience in other minerals, a fanning model is preferred. Extrapolation of the data
to geological timescales suggest that fission tracks in monazite are very sensitive to low
temperature annealing and that significant shortening of tracks occurs even at ambient
surface temperatures (~20°C) and below. Continued shortening of tracks occurs at
temperatures between ~50 - 160°C when extrapolated to geological timescales, with few
tracks being recorded at lengths of $l/l_0$ <~0.5. Closure temperatures for fission track retention
in monazite are estimated to be only 46 - 25°C over geological timescales of $10^6 – 10^7$ years,
consistent with the <50°C estimate of Weise et al. (2009).

As highlighted in Laslett et al. (1987), there is no good reason why the contours in the fanning
Arrhenius plot need to be straight and an alternative fanning curvilinear model has been




proposed in the case of apatite by Ketcham et al. (2007, 1999). Further experiments to
increase the number of data points, especially for even longer heating schedules, would be
required to test this model in monazite. Factors that have not been considered in this study
and could possibly influence annealing kinetics are compositional effects (e.g. Green et al.,
1985), radiation damage effects on etching (e.g. Gleadow, 1981) and radiation enhanced
annealing (e.g. McDannell et al. 2019). The validity of this study still requires further
confirmation by comparing the predictions from our laboratory results with observations
from natural field examples and borehole studies.  Nevertheless, it is clear that fission tracks
in monazite have the lowest thermal stability of any mineral so far studied and this system
has potential for use as an ultra-low temperature thermochronometer.

***Author Contributions***
SJ is a PhD student who obtained and analysed the presented data as well as prepared the
original manuscript. AG and BK provided supervision and contributed to several drafts of the
original manuscript. Sample material was provided by AG and BK.

***Competing Interests***
The authors declare that they have no conflict of interest.

***Acknowledgements***
We thank Ling Chung for assistance and advice on sample preparation methods and Cameron
Patrick from The University of Melbourne Statistical Consulting Centre for assistance with
statistical analysis. SJ also acknowledges funding from the Australian Government through an
Australian Postgraduate Award (APA). The Melbourne thermochronology laboratory is
supported by the AuScope Program funded under the National Collaborative Research
Infrastructure Strategy (NCRIS).



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
