# Peer review of "Thermal Annealing of Implanted 252Cf Fission-Tracks in Monazite"

_Geochronology, 2020_

## Referee Comment (RC1) · Ewald Hejl (Referee) · 23 Oct 2020

General comments:

This manuscript is built on high-quality data; it is well organized and conclusive. The topic is relevant for low-temperature chronology of basement rocks, sedimentary rocks, for geomorphology, and because of the very low closure temperature for fission tracks in monazite (Tc < 45 °C) it can be also useful for paleoclimate research.

Therefore, I strongly recommend this manuscript for publication. About 50 % of text and figures can be published in the present form. Some paragraphs and sentences need minor revision. Argumentation is well-founded. I did not discover major shortcomings – even after very attentive reading.

[Figure]

My overall impression is very good.

Specific comments:

My main concern is that the wording "parallel to the c-axis" is a little bit misleading and not precise enough. In contrast to apatite and other minerals with one main symmetry axis (hexagonal, trigonal or tetragonal), monazite crystals are monoclinic and their c-axis is not a symmetry axis. Therefore, any surfaces parallel to this c-axis are not necessarily symmetrically equivalent. In fact, monazite's only symmetry axis is its b-axis. This axis produces a congruent position after a rotation of 180°. It is a binary symmetry axis. The faces with Miller indices (100) and (-100) are symmetrically equivalent. They cannot form a prism but only a pinacoid. The wording in line 13 "... parallel and perpendicular to (100) prismatic faces" must be replaced by "... parallel and perpendicular to (100) pinacoidal faces".

Just for better understanding: Monazite crystallizes in the crystal class monoclinic prismatic (2/m). All faces which are parallel to two axes form pinacoids and occur only two times on the crystal: (100) and (-100); (010) and (0-10); (001) and (00-1). Faces cutting the b-axis and one or two other axes form prisms. Each of them occurs four times on the crystal. (110) and its three symmetrically corresponding faces form a prism. This is also the case for (311), (011), and (111) and their symmetrically corresponding faces. All these prismatic forms comprise four symmetrically corresponding faces.

Track annealing was investigated on surfaces perpendicular to the crystallographic c-axis and on surfaces parallel to both b-axis and c-axis, i.e. on (100) pinacoidal faces. I suggest to replace the wording "parallel to c-axis" or "// c-axis" by either "parallel to b- and c-axes" or simply by "(100)" all over the manuscript. In the third column of Table 3 you could simply write (100) instead of // c-axis. Please note that "perpendicular to c-axis" is precise and correct. This should not be changed because a face with Miller indices (001) is not exactly perpendicular to the c-axis. The angle between a- and the c-axis is not 90°. Only // c-axis should be replaced by (100).

Another aspect is the azimuth of implanted tracks. The sentence in lines 317 to 319 explains that the dip angle (30°) was constant but "there was limited control on the azimuth orientations" of implanted tracks. Eventually this should be stated earlier in the manuscript (2. Experimental methods) and not only in chapter 4 (Discussion). I understand quite well that handling of such small grains is very difficult and nerve-racking and that a precise control on track azimuth orientation is almost impossible. We should bear in mind that some tracks on faces (100) can be almost perpendicular to c-axis and thus may have similar orientations than some of the tracks implanted on faces perpendicular to c-axis. This could be a reason for lower apparent anisotropy of annealing. I expect that anisotropy will become stronger with a better azimuth control.

I fully agree that the higher temperature limit of the MPAZ can be defined at l/l0 = 0,50 (lines 21 and 22; 540 and 541; but I wonder if this is understandable for readers which are not familiar with fission track dating.

The original PAZ concept was formulated by WAGNER (1972) under the name of "partial stability zone" (The geological interpretation of fission track ages, Amer. Nucl. Soc., 15, 117). At this time and for the next ca. 20 years it was generally assumed that radiation damage in latent fission tracks is erased from the track ends inwards (at least for apatite). A fresh apatite fission track has an etchable length of about 16 $\mu$m, and with progressive fading it becomes shorter but never fragmented. Thus, over the whole temperature range of the PAZ there should be a strict proportionality between etchable (confined) track length and areal track density on etched surfaces. The original PAZ concept has predicted that in the middle of the PAZ apatite fission tracks have a length of 8 $\mu$m (= 16/2) and that the observed areal density is exactly half of that produced by fresh tracks with a length of 16 $\mu$m. In the meantime we know that this assumption is wrong – at least for the confined track length.

There is strong evidence that apatite fission tracks have an original etchable length of about 16 $\mu$m and in course of annealing they shorten to a length of about 10 $\mu$m. Afterwards, they become fragmented and simply disappear. In fact, confined tracks with

lengths < 8 $\mu$m are rarely observed because they mainly do not exist. Proportionality between the reductions of areal track density and the etchable track track length does not describe the true behaviour of track annealing (HEJL, 1995, Chem. Geol. (Isotope Geoscience Section), 122, 259-269; and several other articles). This statement seems to be true also for monazite. I suggest some additional sentences for a better understanding of the (M)PAZ concept. Otherwise, readers could eventually not understand why 50 % length reduction correspond to almost complete erasure of tracks.

Technical corrections:

Line 244: "However, the same is not true for all of the control measurements" on unannealed samples.

Line 297: Probably Figure 4 instead of Figure 2.

Lines 550 and 551: ". . . the parallel model is slightly preferable". This is in contradiction to line 575 (". . . a fanning model is preferred"). Which model is better or preferred?

Figure 2: Can you eventually indicate the direction of the c-axis in this picture? Just leave it when the azimuth orientation is badly known.

Please also note the supplement to this comment:
https://gchron.copernicus.org/preprints/gchron-2020-24/gchron-2020-24-RC1-supplement.pdf
* * *

---

## Referee Comment (RC2) · Dale Issler (Referee) · 26 Oct 2020

Overview

My perspective on this manuscript is from a person who develops and applies models rather than one who does hands on laboratory work. In my opinion, this a high quality manuscript that presents new and important annealing experimental data that help to constrain a potential new ultra-low temperature thermochronometer using fission tracks in monazite. This study is a nice follow up to previous annealing study of monazite by Weise et al. (2009). The manuscript is well organized and well written and includes essential figures and tables that are required for understanding how the work was done and how the data are used to estimate monazite-annealing temperatures

through geological time. This work is exciting and interesting because these results support the notion that monazite has greater sensitivity at much lower temperatures than other widely used thermochronometers. This can lead to new applications in the earth sciences and perhaps allow for the resolution of previously undetectable thermal events. I recommend that this manuscript be published after some minor revision.

Specific comments

Although the manuscript is efficiently written and easy to follow, I believe that more details on aspects of the experimental methods could be included, probably in a supplementary appendix. The authors give many useful details on track implantation and measurement and that part is fine. We know that accurate laboratory temperatures are critical for calibrating annealing models and that extrapolation of annealing temperatures to geological time scales are particularly sensitive to uncertainties in laboratory temperatures. Therefore, it would be helpful to have some more information on laboratory procedures and conditions. For example, there is no mention of any pre-heating of samples to pre-anneal fossil tracks and eliminate any potential radiation damage. A very brief description is given for the heating apparatus and temperature uncertainties were estimated to be $\pm 2°C$. How was this estimated? Any steps taken to ensure constant isothermal conditions within the heating apparatus where the grains were inserted would be worth noting. This information is important because, for example, differences in the results of fission track annealing experiments for apatite among different laboratories have been attributed to temperature uncertainties.

There is significant variation in the initial track lengths for the control apatites used in the annealing experiments which is attributed to ambient annealing following track implantation. Based on the results in Table 1, I assume there is too little compositional variability among the grains to have a significant affect, and if spontaneous tracks were pre-annealed, then any contribution from potential radiation damage related to variable Th and U concentration is unlikely as well. Ambient annealing seems like a reasonable hypothesis so it is unfortunate that there is no information on the time between track

implantation and etching. It would be useful in the supplement to provide some information concerning the order of steps that were performed so that readers could get an idea of the relative time scales involved. How were the data acquired? Were the control grains etched before or after the annealing experiments and did they proceed in some specific order. Presumably, the last ones to be etched had more time for ambient annealing. Did the experiments proceed in the order of shortest heating duration to longest heating duration? Was etching done at the end of each experiment or after all experiments were finished. If this information is available, it may be helpful for a better understanding of the results.

The authors point out that the results in their Figure 6 suggest that anisotropic annealing increases the standard deviation at short mean track lengths, similar to confined tracks in apatite. However, unlike apatite, the standard deviations are also larger at long lengths and this may be attributed to ambient annealing affecting the longer tracks. The U shape distribution of points seems clear in the figure and is worth discussing. There seems to be a slight hint of this pattern in the residual plots in Figure 9.

The last paragraph in the conclusions seems to be more appropriate for the discussion section. The discussion section could be expanded to elaborate on some of the recommendations for future work and the possible influences of other factors on monazite fission track annealing. For example, in addition to using longer heating schedules, low temperature geological benchmarks would be needed to help constrain a fanning curvilinear model like the one used by Ketcham et al. (1999, 2007). That model seems to be better suited to accounting for low temperature annealing of apatite fission tracks. Given the apparent very low annealing temperatures inferred for monazite, it could be very difficult to constrain ambient annealing in a model without such data.

In general, elemental composition has been neglected in many apatite fission-track thermochronology studies and therefore the full potential of multikinetic annealing behaviour has not been utilized. My hope is that this does not happen with monazite should composition turn out to be an important factor influencing annealing. Chemically heterogeneous apatite is widespread in detrital apatite of all ages and it is common for two or three (and sometimes more) kinetic populations to be present in a sample. I have ~200 detrital multikinetic apatite FT samples, some of which contain kinetic populations with differences in annealing temperatures that can approach 100°C, in general agreement with temperature ranges inferred from models calibrated using the results of annealing experiments (Ketcham et al., 1999, 2007). If monazite composition is important, then it may shift the annealing range to higher temperatures and allow for a better-calibrated annealing model. Otherwise, it may difficult to calibrate a model where tracks are unstable at temperatures of geological interest.

Although not stated in the manuscript, it seems that etching is nearly isotropic in monazite or, at least, far less anisotropic than apatite. In addition, in the conclusions it is stated that anisotropic annealing occurs at measurably different rates with respect to the crystallographic axes in monazite. It would be worth mentioning the degree of anisotropy with respect to a well-known mineral like apatite as a reference for readers. Elsewhere in manuscript, the degree of anisotropy is not considered to be that large and that point could be made in the conclusions as well.

Technical Corrections

A few minor technical corrections are needed.

1) Lines 64,and 65, page 3. There seems to be a paragraph break but no space between these lines. 2) Line 264, page 13. Replace "a" with "an" before unannealed. 3) Line 297, page 14. Reference should be to Figure 4, not Figure 2. 4) Line 393-394, page 19. At first, it seemed odd to be showing the double Box-Cox transform when only the single form was being used here. Later in the text, it is mentioned that both are used so it makes sense to include it. However, you might want to modify the description to say you use the term in parentheses in equation 4 when referring to the single Box-Cox transformation. 5) Line 423, page 20. A square bracket is missing at the end of equation 7. 6) Line 502, page 26. Reference should be to Figure 9, not

Figure 7. 7) Line 680, page 32. Reference to the third author should be O'Sullivan, P.B. Also, a space should come after the colon, not before it.

Reviewer: Dale Issler

---

## Author Comment (AC1) · 4 Dec 2020

We would like to thank Ewald Hejl for his constructive comments to improve the paper for the journal "Geochronology". Please see below our responses to all suggestions and comments.

Referee 1 (Ewald Hejl)

General comments: This manuscript is built on high-quality data; it is well organized and conclusive. The topic is relevant for low-temperature chronology of basement rocks, sedimentary rocks, for geomorphology, and because of the very low closure temperature for fission tracks in monazite (Tc < 45ïĊřC) it can be also useful for paleoclimate research. Therefore, I strongly recommend this manuscript for publication.

[Figure]

About 50% of text and figures can be published in the present form. Some paragraphs and sentences need minor revision. Argumentation is well-founded. I did not discover major shortcomings – even after very attentive reading.

My overall impression is very good.

Specific comments: My main concern is that the wording "parallel to the c-axis" is a little bit misleading and not precise enough. In contrast to apatite and other minerals with one main symmetry axis (hexagonal, trigonal or tetragonal), monazite crystals are monoclinic and their c-axis is not a symmetry axis. Therefore, any surfaces parallel to this c-axis are not necessarily symmetrically equivalent. In fact, monazite's only symmetry axis is its b-axis. This axis produces a congruent position after a rotation of 180_. It is a binary symmetry axis. The faces with Miller indices (100) and (-100) are symmetrically equivalent. They cannot form a prism but only a pinacoid. The wording in line 13 ". . . parallel and perpendicular to (100) prismatic faces" must be replaced by ". . . parallel and perpendicular to (100) pinacoidal faces".

Just for better understanding: Monazite crystallizes in the crystal class monoclinic prismatic (2/m). All faces which are parallel to two axes form pinacoids and occur only two times on the crystal: (100) and (-100); (010) and (0-10); (001) and (00-1). Faces cutting the b-axis and one or two other axes form prisms. Each of them occurs four times on the crystal. (110) and its three symmetrically corresponding faces form a prism. This is also the case for (311), (011), and (111) and their symmetrically corresponding faces. All these prismatic forms comprise four symmetrically corresponding faces.

Track annealing was investigated on surfaces perpendicular to the crystallographic c-axis and on surfaces parallel to both b-axis and c-axis, i.e. on (100) pinacoidal faces. I suggest to replace the wording "parallel to c-axis" or "// c-axis" by either "parallel to band c-axes" or simply by "(100)" all over the manuscript. In the third column of Table 3 you could simply write (100) instead of // c-axis. Please note that "perpendicular to c-axis" is precise and correct. This should not be changed because a face with Miller

indices (001) is not exactly perpendicular to the c-axis. The angle between a- and the c-axis is not 90_. Only // c-axis should be replaced by (100).

RESPONSE: We agree with this and will make the appropriate changes throughout the text.

Another aspect is the azimuth of implanted tracks. The sentence in lines 317 to 319 explains that the dip angle (30ïĆř) was constant but "there was limited control on the azimuth orientations" of implanted tracks. Eventually this should be stated earlier in the manuscript (2. Experimental methods) and not only in chapter 4 (Discussion). I understand quite well that handling of such small grains is very difficult and nerverack-ing and that a precise control on track azimuth orientation is almost impossible. We should bear in mind that some tracks on faces (100) can be almost perpendicular to c-axis and thus may have similar orientations than some of the tracks implanted on faces perpendicular to c-axis. This could be a reason for lower apparent anisotropy of annealing. I expect that anisotropy will become stronger with a better azimuth control.

RESPONSE: Yes, this is quite correct, handling of the small size of the grains can make controlling track azimuth very difficult. We will therefore add additional detail to the text explaining that we did our best to control track azimuth in 2. Experiments Methods.

I fully agree that the higher temperature limit of the MPAZ can be defined at l/l0 = 0,50 (lines 21 and 22; 540 and 541; but I wonder if this is understandable for readers which are not familiar with fission track dating.

The original PAZ concept was formulated by WAGNER (1972) under the name of "par-tial stability zone" (The geological interpretation of fission track ages, Amer. Nucl. Soc., 15, 117). At this time and for the next ca. 20 years it was generally assumed that ra-diation damage in latent fission tracks is erased from the track ends inwards (at least for apatite). A fresh apatite fission track has an etchable length of about 16 ïA■m, and with progressive fading it becomes shorter but never fragmented. Thus, over the whole temperature range of the PAZ there should be a strict proportionality between etchable

(confined) track length and areal track density on etched surfaces. The original PAZ concept has predicted that in the middle of the PAZ apatite fission tracks have a length of 8 ïA▪m (= 16/2) and that the observed areal density is exactly half of that produced by fresh tracks with a length of 16 ïA▪m. In the meantime we know that this assumption is wrong – at least for the confined track length.

There is strong evidence that apatite fission tracks have an original etchable length of about 16 ïA▪m and in course of annealing they shorten to a length of about 10 ïA▪m. Afterwards, they become fragmented and simply disappear. In fact, confined tracks with lengths < 8 ïA▪m are rarely observed because they mainly do not exist. Proportionality between the reductions of areal track density and the etchable track length does not describe the true behaviour of track annealing (HEJL, 1995, Chem. Geol. (Isotope Geoscience Section), 122, 259-269; and several other articles). This statement seems to be true also for monazite. I suggest some additional sentences for a better understanding of the (M)PAZ concept. Otherwise, readers could eventually not understand why 50 % length reduction correspond to almost complete erasure of tracks.

RESPONSE: Noted. We will include some additional sentences explaining how the closure temperature concept is applied to fission track annealing.

Technical corrections: Line 244: "However, the same is not true for all of the control measurements" on unannealed samples.

RESPONSE: Will change this accordingly.

Line 297: Probably Figure 4 instead of Figure 2.

RESPONSE: Will change this accordingly.

Lines 550 and 551: ": : : the parallel model is slightly preferable". This is in contradiction to line 575 (": : : a fanning model is preferred"). Which model is better or preferred?

[Figure]

RESPONSE: Lines 550 and 551 state that based on the model statistics (coefficients of determination), the parallel model is slightly preferable, but the difference is negligible, as noted in Lines 574-575, leaving both models viable. By analogy with annealing studies in zircon and apatite that show fanning models best fit their respective datasets, we conclude that the fanning model is preferable for monazite. We will modify the text to make this conclusion more clear.

Figure 2: Can you eventually indicate the direction of the c-axis in this picture? Just leave it when the azimuth orientation is badly known.

RESPONSE: I will find the original image and see if I can identify the c-axis.

Please also note the supplement to this comment:
https://gchron.copernicus.org/preprints/gchron-2020-24/gchron-2020-24-AC1-supplement.pdf

---

## Author Comment (AC2) · 4 Dec 2020

We would like to thank Dale Issler for his constructive comments to improve our paper for the journal "Geochronology". Please see below our responses to all suggestions and comments.

Referee 2 (Dale Issler)

Overview My perspective on this manuscript is from a person who develops and applies models rather than one who does hands on laboratory work. In my opinion, this a high quality manuscript that presents new and important annealing experimental data that help to constrain a potential new ultra-low temperature thermochronometer using fission tracks in monazite. This study is a nice follow up to previous annealing study of

monazite by Weise et al. (2009). The manuscript is well organized and well written and includes essential figures and tables that are required for understanding how the work was done and how the data are used to estimate monazite-annealing temperatures through geological time. This work is exciting and interesting because these results support the notion that monazite has greater sensitivity at much lower temperatures than other widely used thermochronometers. This can lead to new applications in the earth sciences and perhaps allow for the resolution of previously undetectable thermal events. I recommend that this manuscript be published after some minor revision.

Specific comments Although the manuscript is efficiently written and easy to follow, I believe that more details on aspects of the experimental methods could be included, probably in a supplementary appendix. The authors give many useful details on track implantation and measurement and that part is fine. We know that accurate laboratory temperatures are critical for calibrating annealing models and that extrapolation of annealing temperatures to geological time scales are particularly sensitive to uncertainties in laboratory temperatures. Therefore, it would be helpful to have some more information on laboratory procedures and conditions. For example, there is no mention of any pre-heating of samples to pre-anneal fossil tracks and eliminate any potential radiation damage. A very brief description is given for the heating apparatus and temperature uncertainties were estimated to be ∼2ïĆřC. How was this estimated? Any steps taken to ensure constant isothermal conditions within the heating apparatus where the grains were inserted would be worth noting. This information is important because, for example, differences in the results of fission track annealing experiments for apatite among different laboratories have been attributed to temperature uncertainties. There is significant variation in the initial track lengths for the control apatites used in the annealing experiments which is attributed to ambient annealing following track implantation. Based on the results in Table 1, I assume there is too little compositional variability among the grains to have a significant affect, and if spontaneous tracks were pre-annealed, then any contribution from potential radiation damage related to variable Th and U concentration is unlikely as well. Ambient annealing seems like a reasonable

hypothesis so it is unfortunate that there is no information on the time between track implantation and etching. It would be useful in the supplement to provide some information concerning the order of steps that were performed so that readers could get an idea of the relative time scales involved. How were the data acquired? Were the control grains etched before or after the annealing experiments and did they proceed in some specific order. Presumably, the last ones to be etched had more time for ambient annealing. Did the experiments proceed in the order of shortest heating duration to longest heating duration? Was etching done at the end of each experiment or after all experiments were finished. If this information is available, it may be helpful for a better understanding of the results.

RESPONSE: We will happily add pre-annealing conditions for the monazite specimens in the main body of the text. As requested, we will add a supplementary information section at the end of the manuscript. This will outline the experimental procedures in further detail. Temperatures in the Aluminium heating block were monitored using a digital thermometer, cross calibrated against a mercury thermometer and checked as often as possible. This is stated in text: "The block heater was covered by a ceramic foam block for insulation through which a probe could be inserted to monitor temperatures". The ceramic foam block helped to keep temperatures constant within the heating block. Previous experiments have shown that there is no detectable temperature variation across all the wells in the block heater.

The authors point out that the results in their Figure 6 suggest that anisotropic annealing increases the standard deviation at short mean track lengths, similar to confined tracks in apatite. However, unlike apatite, the standard deviations are also larger at long lengths and this may be attributed to ambient annealing affecting the longer tracks. The U shape distribution of points seems clear in the figure and is worth discussing. There seems to be a slight hint of this pattern in the residual plots in Figure 9. The last paragraph in the conclusions seems to be more appropriate for the discussion section. The discussion section could be expanded to elaborate on some of the recommendations

for future work and the possible influences of other factors on monazite fission track annealing. For example, in addition to using longer heating schedules, low temperature geological benchmarks would be needed to help constrain a fanning curvilinear model like the one used by Ketcham et al. (1999, 2007). That model seems to be better suited to accounting for low temperature annealing of apatite fission tracks. Given the apparent very low annealing temperatures inferred for monazite, it could be very difficult to constrain ambient annealing in a model without such data. In general, elemental composition has been neglected in many apatite fission-track thermochronology studies and therefore the full potential of multikinetic annealing behaviour has not been utilized. My hope is that this does not happen with monazite should composition turn out to be an important factor influencing annealing. Chemically heterogeneous apatite is widespread in detrital apatite of all ages and it is common for two or three (and sometimes more) kinetic populations to be present in a sample. I have ∼200 detrital multikinetic apatite FT samples, some of which contain kinetic populations with differences in annealing temperatures that can approach 100ïĆřC, in general agreement with temperature ranges inferred from models calibrated using the results of annealing experiments (Ketcham et al., 1999, 2007). If monazite composition is important, then it may shift the annealing range to higher temperatures and allow for a better-calibrated annealing model. Otherwise, it may difficult to calibrate a model where tracks are unstable at temperatures of geological interest.

RESPONSE: We agree with the higher standard deviations on the longer track lengths being possibly attributed to ambient annealing. We will add a couple sentences addressing this.

As requested, we will move the last paragraph to Sections 5 and 6. Here we will elaborate on future work using the model of Ketcham et al. (1999, 2007), which could also be useful to constrain ambient temperature annealing in monazite. We will also expand our discussion with extra sentences about the factors that could affect fission track annealing in monazite. While most monazite specimens are reported to be Cedominated, it will still be important to characterize any monazite used for future annealing experiments and case studies. Monazite can be compositionally zoned, so I would expect there to be notable compositional differences between and within grains in any one sample.

Although not stated in the manuscript, it seems that etching is nearly isotropic in monazite or, at least, far less anisotropic than apatite. In addition, in the conclusions it is stated that anisotropic annealing occurs at measurably different rates with respect to the crystallographic axes in monazite. It would be worth mentioning the degree of anisotropy with respect to a well-known mineral like apatite as a reference for readers. Elsewhere in manuscript, the degree of anisotropy is not considered to be that large and that point could be made in the conclusions as well.

RESPONSE: It is true that track etching appears to be more isotropic than in apatite – from our track diameter data in Table 2 – and, although there is some detectable annealing anisotropy, this is also much less than seen in apatite. We will add a comment to this effect as suggested.

A few minor technical corrections are needed. 1) Lines 64, and 65, page 3. There seems to be a paragraph break but no space between these lines.

RESPONSE: Thanks for picking this up, we will fix it.

2) Line 264, page 13. Replace "a" with "an" before unannealed.

RESPONSE: We will correct this.

3) Line 297, page 14. Reference should be to Figure 4, not Figure 2.

RESPONSE: Will correct this.

4) Line 393-394, page 19. At first, it seemed odd to be showing the double Box-Cox transform when only the single form was being used here. Later in the text, it is mentioned that both are used so it makes sense to include it. However, you might want

to modify the description to say you use the term in parentheses in equation 4 when referring to the single Box-Cox transformation.

RESPONSE: Will address this.

5) Line 423, page 20. A square bracket is missing at the end of equation 7.

RESPONSE: Will correct this.

6) Line 502, page 26. Reference should be to Figure 9, not Figure 7.

RESPONSE: Will correct this.

7) Line 680, page 32. Reference to the third author should be O'Sullivan, P.B. Also, a space should come after the colon, not before it.

RESPONSE: Will correct this.

Please also note the supplement to this comment:
https://gchron.copernicus.org/preprints/gchron-2020-24/gchron-2020-24-AC2-supplement.pdf